# Learning Control for Flexible Manipulators with Varying Loads: A Composite Method with Robust Adaptive Dynamic Programming and Robust Sliding Mode Control

**Yiming Xu [1], Xiao Wang [2], Li Wang [1], Kai Wang [1] and Lei Ma [2,*]**

1   Jiangsu Institute of Automation, Lianyungang 222000, China; xuyiming@jari.cn (Y.X.); wangli1218@seu.edu.cn (L.W.); wangkai@jari.cn (K.W.)
2   School of Information and Control Engineering, China University of Mining and Technology, Xuzhou 221000, China; xiaowang@cumt.edu.cn
*   Correspondence: malei@cumt.edu.cn

**Abstract:** This paper focuses on the learning-based motion control for flexible manipulators with varying loads via the singularly perturbed technique. Considering the two-timescale feature of the flexible manipulator, system dynamics are decomposed into fast and slow subsystems, and corresponding sub-controllers are designed with robust adaptive dynamic programming (RADP) and robust sliding mode control (RSMC) methods, respectively. In the proposed composite control framework, an RADP-based sub-controller is developed to realize the trajectory tracking and alleviate the parametric uncertainty utilizing rotating angles in the slow timescale, while an RSMC sub-controller is introduced to improve the vibration suppression in the fast timescale. Finally, the stability of the closed-loop system is guaranteed, and simulations are carried out to show the effectiveness of the proposed control algorithm.

**Keywords:** flexible manipulator; singular perturbation theory; adaptive dynamic programming; sliding-mode control; varying loads

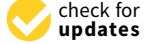



## 1. Introduction

Over the past few decades, flexible manipulators have played an important role in robotics, and have received much attention for their advantages, such as fast speed, low power consumption, etc. [1,2]. Unlike rigid manipulators, flexible manipulators are known to possess uncertainty dynamics with infinite order, rigid–flexible coupling, and nonlinear dynamics. These attributes lead to significant challenges for the position control because it is difficult to precisely track the desired position and suppress the vibration caused by flexibility simultaneously [3,4].

The motion dynamics of flexible manipulators include macro position rotating (slow dynamics) and micro elastic vibration (fast dynamics). Many control technologies have been proposed to manage these dynamics, involving PID control, sliding mode control, fuzzy control, neural network control, etc. [5–9]. Classical adaptive and neural network controllers have been proposed in [10,11]. Nevertheless, the aforementioned controllers are directly designed by dealing with the coupled macro and micro dynamics together, which usually operate on different timescales (also called the two-timescale feature, TTSF). As such, inaccurate control is caused and ill-conditioned numerical issues will occur by the direct application of the traditional methods to the flexible manipulators due to the TTSF. To solve such difficulties, singular perturbation (SP) theory is introduced in the motion control for flexible manipulators.

By using the SP theory, the state-space model of the flexible manipulators can be decomposed into two low-order subsystems, namely the slow subsystem in the slow timescale and the fast subsystem in the fast timescale. Then, sub-controllers are designed in

slow and fast timescales in correspondence to slow and fast subsystems [12,13], and fruitful results have been achieved. Based on the decomposed subsystems of flexible manipulators, a composite learning controller using neural network control and a disturbance observer was designed to improve the tracking precision [14]. In [15], the fuzzy sliding mode control method was used to design a slow subsystem controller, and the LQR control method was developed for the fast subsystem controller design. In [16], state feedback control was proposed, and trajectory tracking and vibration suppression were realized simultaneously during movement. These examples show the high accuracy of positional tracking and fast vibration suppression using the SP theory. In [17], a composite controller based on computed torque control and linear-quadratic control is designed, which suppresses the joint and link vibration satisfactorily, and great trajectory tracking performance is achieved. Though research results have been obtained for the flexible manipulators with fixed loads, when it comes to the varying loads, the aforementioned algorithms are no longer applicable due to the existence of vibration in the movement and imprecise parameters.

To cope with such problems, in this paper, a composite learning controller is proposed with robust adaptive dynamic programming (RADP) and robust sliding-mode control (RSMC) methods. A slow sub-controller based on RADP is developed to realize trajectory tracking using rotating angles with no knowledge of flexible manipulators. Then, a fast sub-controller based on RSMC is designed, taking the dynamic uncertainties into account. Finally, a composite controller based on RADP and RSMC is formulated, and the stability is proven via the SP theory. Simulation results of different varying loads verify the accuracy of the decomposition and the effectiveness of the controller.

Contributions of this paper are summarized as follows:

(1) A novel learning-based composite controller is proposed, for the first time, for flexible manipulators with RADP and RSMC;
(2) The possible ill-conditioned numerical is avoided and the stability of the system is guaranteed via the proposed SP-based control algorithm;
(3) Varying loads and parameters, as the first attempt, are taken into consideration in two timescales, which significantly improves the positioning accuracy.

## 2. Dynamics Modeling and Decomposition

The dynamics of series flexible manipulators with multi-degrees of freedom is established using Lagrange and assumed-mode methods as follows:

$$M \begin{bmatrix} \ddot{\theta} \\ \ddot{q} \end{bmatrix} + \begin{bmatrix} 0 & 0 \\ 0 & D \end{bmatrix} \begin{bmatrix} \theta \\ q \end{bmatrix} + \begin{bmatrix} S_1 \\ S_2 \end{bmatrix} = \begin{bmatrix} 1 \\ 0 \end{bmatrix} \tau \tag{1}$$

where $M = \begin{bmatrix} M_{11} & M_{12} \\ M_{21} & M_{22} \end{bmatrix}$ is the inertia matrix, which is positive definite, non-singular, and symmetric, $D$ is the stiffness matrix, and $S_1, S_2$ are the nonlinear terms of Coriolis and centrifugal forces, where $S_1 = 0$. The dynamics of flexible manipulators can be modeled as a two-timescale system with coupled slow and fast dynamics, which can be seen from [13].

The variables $\tau \in R^{1 \times 1}$ are control torque; $\theta \in R^{n \times 1}$ and $q \in R^{nm \times 1}$ are defined, respectively, as:

$$\theta = \begin{bmatrix} \theta_1 & \dots & \theta_n \end{bmatrix}^T$$

$$q = \begin{bmatrix} q_{11} & \dots & q_{1m} & \dots & q_{n1} & \dots & q_{nm} \end{bmatrix}^T$$

$\theta_i$ describes the *ith* joint rotating angles and $q_{ij}$ describes the *ith* manipulator and *jth* vibration modes. Let $H$ be the inverse matrix of $M$; thus,

$$H = M^{-1} = \begin{bmatrix} H_{11} & H_{12} \\ H_{21} & H_{22} \end{bmatrix}$$

Then, multiplying both sides of dynamics (1) with $H$ yields:

$$\begin{cases} \ddot{\theta} = -H_{11}(S_1 - \tau) - H_{12}(S_2 + Dq) \\ \ddot{q} = -H_{21}(S_1 - \tau) - H_{22}(S_2 + Dq) \end{cases} \tag{2}$$

Based on SP theory, multi-timescale factor $\varepsilon = 1/d$ is introduced, where $d$ is the minimum eigenvalue of $D$. On this basis, new variables are defined as $z = q/\varepsilon$ and $\tilde{D} = \varepsilon D$. Then, (2) can be rewritten as:

$$\begin{cases} \ddot{\theta} = -H_{11}(S_1 - \tau) - H_{12}(S_2 + \tilde{D}z) \\ \varepsilon \ddot{z} = -H_{21}(S_1 - \tau) - H_{22}(S_2 + \tilde{D}z) \end{cases} \tag{3}$$

Letting $\varepsilon = 0$, $z$ in (3) can be solved as:

$$z^s = \tilde{D}^{-1} H_{22}^{s-1}(-H_{21}^s S_1^s - H_{22}^s S_2^s + H_{21}^s \tau^s) \tag{4}$$

where superscript s denotes slow dynamics, $\theta^s$ is the approximation of $\theta$, and $\tau^s$ is the controller in slow timescale. By substituting (4) into (3), the slow subsystem can be derived as:

$$\ddot{\theta}^s = G\tau^s + GS_1^s \tag{5}$$

where $G = H_{11}^s - H_{12}^s(H_{22}^s)^{-1}H_{21}^s$. Define $\eta_1^s = \theta^s$ and $\eta_2^s = \dot{\theta}^s$. In the slow timescale, the transformed dynamics of (5) can be obtained as:

$$\begin{cases} \dot{\eta}_1^s = \eta_2^s \\ \dot{\eta}_2^s = G\tau^s - GS_1^s \end{cases} \tag{6}$$

To derive the fast subsystem, new variables are defined as $\xi = t/\sqrt{\varepsilon}$ and $z^f = [z_1^f, z_2^f]^T = [z - z^s, \sqrt{\varepsilon}\dot{z}]^T$. In the fast timescale, slow variables are regarded as constants [12], and yield:

$$\begin{aligned} d\theta^s/d\xi = d^2\theta^s/d\xi^2 = 0 \\ dz^s/d\xi = d^2z^s/d\xi^2 = 0 \end{aligned} \tag{7}$$

In the boundary layer of flexible manipulators, by setting $\varepsilon = 0$, the transformed dynamics of the fast subsystem can be obtained as:

$$\begin{aligned} \frac{dz_1^f}{d\xi} = z_2^f \\ \frac{dz_2^f}{d\xi} = -H_{22}^s \tilde{D} z_1^f + H_{21}^s \tau^f \end{aligned} \tag{8}$$

According to Tikhonov's theorem [18], the following relations hold:

$$\begin{aligned} \theta = \theta^s + O(\varepsilon) \\ q = \varepsilon(z^s + z^f) + O(\varepsilon) \end{aligned} \tag{9}$$

In (9), $\theta$ is the high-order infinitesimal of $\theta^s$ and $q$ is the high-order infinitesimal of $\varepsilon(z^s + z^f)$ regarding $\varepsilon$.

## 3. Controller Design

### 3.1. Slow Controller Design

As shown in (5), the trajectory tracking error is defined as:

$$\begin{aligned} e_1 = \theta - \theta_d \\ e_2 = \dot{\theta} - \dot{\theta}_d \end{aligned} \tag{10}$$

where $\theta_d$ denotes the ideal tracking. Define $x_1 = e_1$, $x_2 = e_2$. Thus, the second-order differential ideal position is that $\ddot{\theta}_d = 0$. Define $x_1^s = e_1$, $x_2^s = e_2$. Then, $\dot{x}_1^s = x_2^s$, $\dot{x}_2^s = \ddot{\theta} - \ddot{\theta}_d$. Taking formula (6) into account, formula (11) can be obtained

The slow dynamics can be rewritten as:

$$\dot{x}^s = A^s x^s + B^s \tau^s \tag{11}$$

where $A^s = \begin{bmatrix} 0 & 1 \\ 0 & 0 \end{bmatrix}$, $B^s = \begin{bmatrix} 0 \\ -H_{12}^s(H_{22}^s)^{-1}H_{21}^s + H_{11}^s \end{bmatrix}$.

Based on the LQR method [19], the control goal is to find the optimal law as follows:

$$\tau* = -K * x^s \tag{12}$$

which minimizes the following weighted function:

$$J = \int_0^\infty (x^s)^T Q x^s + (\tau^s)^T R \tau^s dt, \quad x^s(0) = x^s{}_0 \in R^n \tag{13}$$

where $Q$, $R$ are positive and definite. $(A^s, B^s)$ is stabilizable and $(A^s, Q^{1/2})$ is observable. The aim is to solve the following algebraic *Riccati* equation, where $A^s$ and $B^s$ are known:

$$(A^s)^T P + P A^s + Q - P B^s R^{-1}(B^s)^T P = 0 \tag{14}$$

The optimal law is determined using (15), which is not relative to the initial condition:

$$\tau^{s*} = -R^{-1}(B^s)^T P * x^s \tag{15}$$

where $P*$ is the unique solution to the algebraic *Riccati* equation (14).

**Proof.** Define the *Lyapunov* function as:

$$V = (x^s)^T P x^s \tag{16}$$

Differentiating (16) and substituting (11), (14), (15) into it provides:

$$\begin{aligned} \dot{V} &= (\dot{x}^s)^T P x^s + (x^s)^T P \dot{x}^s \\ &= (Ax^s + B\tau^s)^T P x^s + (x^s)^T P(Ax + B\tau^s) \\ &= (x^s)^T (A - BR^{-1}B^T P)^T P(x^s) + (x^s)^T P(A - BR^{-1}B^T P)(x^s) \\ &= -(x^s)^T (-A^T P - PA + PBR^{-1}BP + P^T(BR^{-1}B^T)^T P)(x^s) \\ &= -(x^s)^T (Q + P^T(BR^{-1}B^T)^T P) x^s \end{aligned} \tag{17}$$

where matrix $R$, $P$ are symmetric and positive definite, $Q$ is also positive definite, and we can obtain:

$$\dot{V} < 0 \tag{18}$$

Under the action of controller (15), the slow subsystem is stable. The controller design relies on the accurate parameters $A^s$ and $B^s$, but they are not always measured accurately with the existence of vibration caused by the inertial effect and actuation.

Let $K_0$ be the initial feedback gain matrix and $P_k$ be the solution to the following *Lyapunov* Equation (19):

$$(A^s - BK_k)^T P_k + P_k(A^s - BK_k) + Q + K_k^T R K_k = 0 \tag{19}$$

$$K_k = R^{-1}(B^s)^T P_{k-1} \tag{20}$$

The solution to (14) can be approximated by iteratively updating (19) and (20) [20].

Based on SP theory, the dynamics of flexible manipulators can be regarded as rigidbody, which are linear and controllable. In [21], an RADP algorithm is proposed. The slow dynamics in (11) can be rewritten as:

$$\dot{x}^s = A_k^s x^s + B^s (K_k x^s + \tau^s) \tag{21}$$

where $A_k^s = A^s - B^s K_k$.

Then, we can obtain:

$$
\begin{aligned}
&x^s(t + \Delta t)^T P_k x^s(t + \Delta t) - x^s(t)^T P_k x^s(t) \\
&= \int_t^{t+\Delta t} [(x^s)^T ((A_k^s)^T P_k + P_k A_k^s)) x^s + 2(K_k x^s + \tau^s)^T (B^s)^T P_k x^s] d\sigma \\
&= -\int_t^{t+\Delta t} (x^s)^T Q_k x^s d\sigma + 2\int_t^{t+\Delta t} (K_k x^s + \tau^s)^T R K_{k+1} x^s d\sigma
\end{aligned}
\tag{22}
$$

where $Q_k = Q + K_k^T R K_k$.

In (22), the terms $(A_k^s)^T P_k + P_k A_k^s$ and $(B^s)^T P_k$ relative to $A^s$, $B^s$ are replaced by $-Q_k$ and $R K_{k+1} x^s$, respectively. Then, the optimal control law (12) can be designed by the state of the flexible manipulator dynamics. The following is the execution process of RADP. □

Optimal control laws based on RADP [21].

**Step 1:** Employ $\tau^s = -K_0 x^s + \gamma$ as the initial control law of the slow dynamics, where $\gamma$ is the exploration noise; since the characteristic of vibration exists inevitably for the flexible manipulator system, $\gamma$ is chosen to be 0.

**Step 2:** Compute dynamic matrix $\delta_{x^s x^s}$, $I_{x^s x^s}$, $I_{x^s \tau^s}$ during iteration until they meet the following relations:

$$rank[I_{x^s x^s}, I_{x^s \tau^s}] = \frac{n(n+1)}{2} + mn, \ m = 1, \ n = 2 \tag{23}$$

where

$$
\begin{aligned}
\delta_{x^s x^s} = &[\mu(x^s(t_1)) - \mu(x^s(t_0)), \mu(x^s(t_2)) - \mu(x^s(t_1)), \\
&..., \mu(x^s(t_l)) - \mu(x^s(t_{l-1}))]^T
\end{aligned}
\tag{24}
$$

$$I_{x^s x^s} = \left[ \int_{t_0}^{t_1} x^s \otimes x^s d\tau, \int_{t_1}^{t_2} x^s \otimes x^s d\tau, \ldots, \int_{t_{l-1}}^{t_l} x^s \otimes x^s d\tau \right]^T \tag{25}$$

$$I_{x^s \tau^s} = \left[ \int_{t_0}^{t_1} x^s \otimes \tau^s d\sigma, \int_{t_1}^{t_2} x^s \otimes \tau^s d\sigma, \ldots, \int_{t_{l-1}}^{t_l} x^s \otimes \tau^s d\sigma \right]^T \tag{26}$$

$$0 \le t_0 < t_1 < \cdots < t_l$$

$$\mu(x^s) = [(x_1^s)^2, \ x_1^s x_2^s, \ (x_2^s)^2]^T$$

**Step 3:** Solve $P_k$ and $K_{k+1}$ from (24) during iteration.

$$\Omega_k \begin{bmatrix} \gamma(P_k) \\ vec(K_{k+1}) \end{bmatrix} = \Xi_k \tag{27}$$

where $(x^s)^T Q_k (x^s) = ((x^s)^T \otimes (x^s)^T) vec(Q_k)$, $\Omega_k = [\delta_{x^s x^s} \ -2I_{x^s x^s}(I_n \otimes K_k^T R) - 2I_{x^s \tau^s}(I_n \otimes R)]$, $\Xi_k = -I_{x^s x^s} vec(Q_k)$, $\gamma(P) = [p_{11} \ p_{12} \ ... \ 2p_{1n} \ p_{22} \ p_{23} \ ... \ p_{n-1} \ p_{nn}]^T$, $\Omega_k \in R^{l*[\frac{1}{2}n(n+1)+mn]}$, $\Xi_k \in R^l$

Since $\Omega_k$ has full column rank for all $k \in Z_+$, formula (27) can be solved as:

$$\begin{bmatrix} \gamma(P_k) \\ vec(K_{k+1}) \end{bmatrix} = (\Omega_k^T \Omega_k)^{-1} \Omega_k^T \Xi_k \tag{28}$$

**Step 4:** Solve $P_k$ and $K_{k+1}$ iteratively from (27) until $\|P_k - P_{k-1}\| \leq \alpha, \alpha > 0$; if not, return to Step 3.

**Step 5:** Let $K^* = K_k$, and then the optimal controller for the slow subsystem can be solved as:

$$\tau^s = -K^* x^s \tag{29}$$

**Theorem 1.** *Since $(A^s, (Q^s)^{1/2})$ is observable and $K_0^s$ is any stabilizing feedback gain matrix, the subsystem (11) is asymptotically stable under the optimal control law (29).*

### 3.2. Fast Controller Design

In view of the parameter measurement error, a robust sliding-mode controller is designed to suppress the vibration. In the fast timescale, the dynamics of vibration are rewritten as:

$$\dot{x}^f = A^f x^f + B^f \tau^f + f_d \tag{30}$$

where $A^f = \begin{bmatrix} 0 & I \\ -H_{22}^s \tilde{D} & 0 \end{bmatrix}$, $B^f = \begin{bmatrix} 0 & H_{21}^s \end{bmatrix}^T$, $|f_d| < F$, $F$ is the constant upper bounds.

Design the sliding-mode function as:

$$s = G x^f \tag{31}$$

where $G > 0$.

The fast controller can be designed as:

$$\tau^f = -(GB^f)^{-1}[GA^f x^f + \eta \operatorname{sgn}(s) + GF \operatorname{sgn}(s)] \tag{32}$$

**Proof.** Define the *Lyapunov* function as:

$$V = \frac{1}{2} s \dot{s} \tag{33}$$

□

Taking the formulas (27)–(30) into account, the differential of the above Lyapunov function can be solved as:

$$
\begin{aligned}
\dot{V} &= s\dot{s} \\
&= s\left[G(A^f x^f + B^f \tau^f + f_d)\right] \\
&= sG\left[A^f x^f + B^f(-(GB^f)^{-1}(GA^f x + \eta\operatorname{sgn}(s) + GF\operatorname{sgn}(s))) + f_d\right] \\
&= s[-\eta\operatorname{sgn}(s) - GF\operatorname{sgn}(s) + Gf_d] \\
&= -\eta|s| - G|s|(F - f_d) \\
&< 0, \qquad \forall s \neq 0
\end{aligned}
\tag{34}
$$

### 3.3. Composite Controller Design

Based on SP theory, the composite controller can be solved as:

$$\tau = \tau^s + \tau^f \tag{35}$$

**Theorem 2.** *Based on SP theory, the stabilities of the slow subsystem (11) and fast subsystem (30) are guaranteed by their own controllers (29) and (32); thus, the stability of the whole-order system (1) is guaranteed under the composite controller (35).*

*REMARK 1:* In this paper, we use the singularly perturbed technique (SPT) to deal with the control of flexible manipulators with varying loads. In such a system, the trajec-

tory dynamic is the slow dynamic with position changes, which is coupled with the fast dynamic of vibration suppression. In the framework of SPT, we can deal with slow and fast dynamics in corresponding timescales, which will contribute to more precise control and less conservativeness.

## 4. Simulation and Analysis

To verify the rightness and effectiveness of the dynamic decomposition and the composite controller proposed in this paper, simulation results are given.

### 4.1. Verify the Dynamic Decomposition Based on SP

Based on SP theory, the dynamics of flexible manipulators are decomposed into slow and fast sub-dynamics. The position curves of original system and the subsystem are shown in Figure 1.

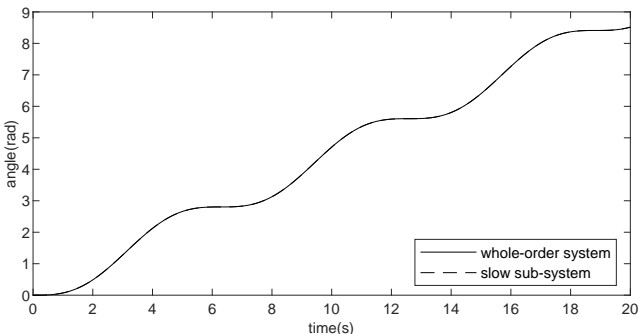

**Figure 1.** Position curves of the original system and the slow subsystem.

### 4.2. Verify the Effectiveness of the Composite Controller

To verify the effectiveness and accuracy of the composite controller (35), control schemes with varying loads are simulated and simulation results are compared with the fuzzy logic composite controller in [12].

In the slow timescale, a slow controller based on RADP by means of $\theta$ is designed to track the angles since $\theta^s \approx \theta$. The initial feedback matrix is chosen as $K_0^s = \begin{bmatrix} 3 & 5 \end{bmatrix}$. The weighted matrixes are set as $Q^s = diag(1, 0.1)$, $R^s = I$.

In the fast timescale, a fast controller in consideration of dynamic uncertainties based on RSMC is designed as shown in (32). The parameters of the controller are chosen as:

$$G = \begin{bmatrix} 0 & 0 & 0.5 & 1 \end{bmatrix}^T$$

$$\eta = 0.5$$

$$F = \begin{bmatrix} 0 & 0 & 50 & 50 \end{bmatrix}^T$$

The simulation results using varying loads, *m* = 0, *m* = 0.1, and *m* = 0.2, are provided in Table 1.

Figures 2 and 3 have shown the second position curves of the origin system and the reduced order system. Figure 4 shows the convergence of $K_k^s$ to its optimal value $K^{sd}$ during the learning process. After finite iterations, the optimal control laws based on RADP and LQR can be solved, and the results are shown in Table 1.

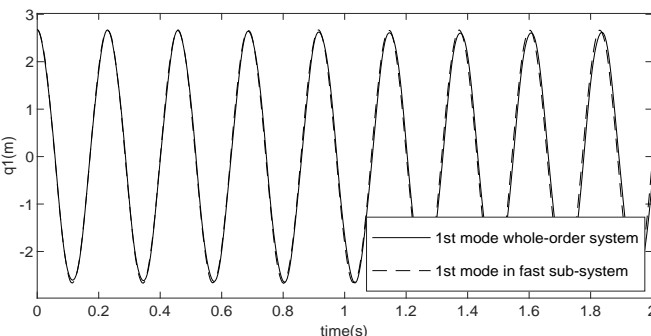

**Figure 2.** $q_1(m)$ of the original system and the fast subsystem.

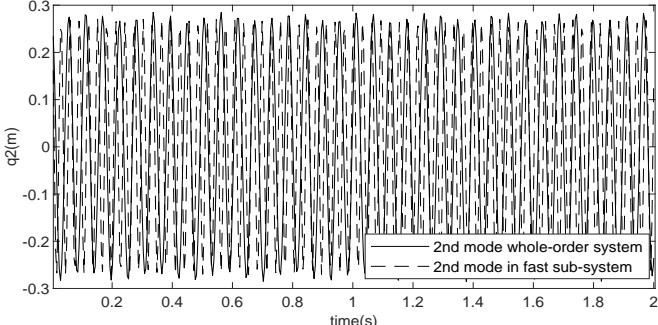

**Figure 3.** $q_2(m)$ mode position curve of the original system and the fast subsystem.

**Table 1.** The optimal control feedback gains with varying loads.

| $m$ | Optimal Laws | The Optimal Control Feedback Gain |
|:---:|:---:|:---:|
| 0 | RADP | $K^{s*} = \begin{bmatrix} 1 & 2.0721 \end{bmatrix}$ |
| | LQR | $K^{sd} = \begin{bmatrix} 1 & 2.0720 \end{bmatrix}$ |
| 0.1 | RADP | $K^{s*} = \begin{bmatrix} 0.9986 & 2.1310 \end{bmatrix}$ |
| | LQR | $K^{sd} = \begin{bmatrix} 1 & 2.1406 \end{bmatrix}$ |
| 0.2 | RADP | $K^{s*} = \begin{bmatrix} 0.9986 & 2.1310 \end{bmatrix}$ |
| | LQR | $K^{sd} = \begin{bmatrix} 1 & 2.1406 \end{bmatrix}$ |

According to Figure 4 and Table 2, $K^{s*} \approx K^{sd}$. The optimal control laws can be solved regardless of the m value chosen under RADP, with no knowledge of the system parameters.

**Table 2.** The optimal control feedback gains with varying loads.

| References | Results |
|:---:|:---:|
| [5] | Full-state tracking PID controller. |
| [14] | A composite learning controller with neural network and disturbance observer. |
| [15] | A composite learning controller with fuzzy sliding mode control and LQR control. |
| [17] | A composite learning controller with computed torque control and linear-quadratic control. |

Figure 5 shows that the controller based on RADP and RSMC has better performance with varying loads. Figures 6 and 7 are the first two mode position curves, which show that the vibration is rapidly suppressed under the controller.

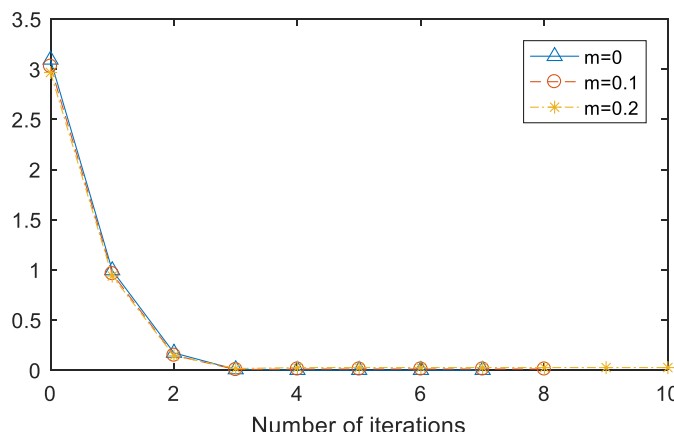

**Figure 4.** Convergence of $K_k^s$ to its optimal value $K^{sd}$ during the learning process.

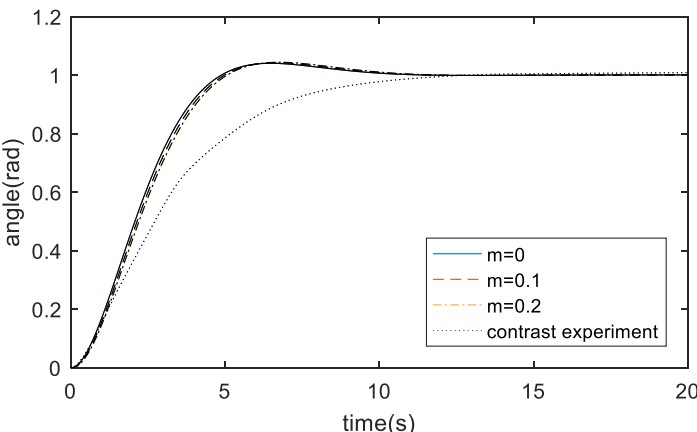

**Figure 5.** The position curve of flexible manipulators.

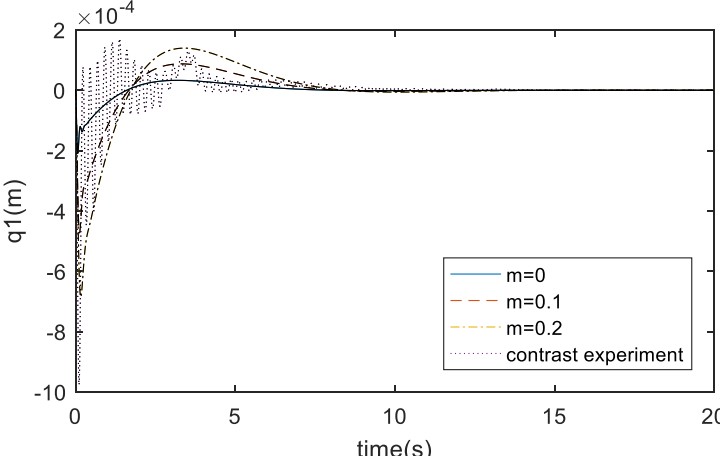

**Figure 6.** The first mode vibration of flexible manipulators.

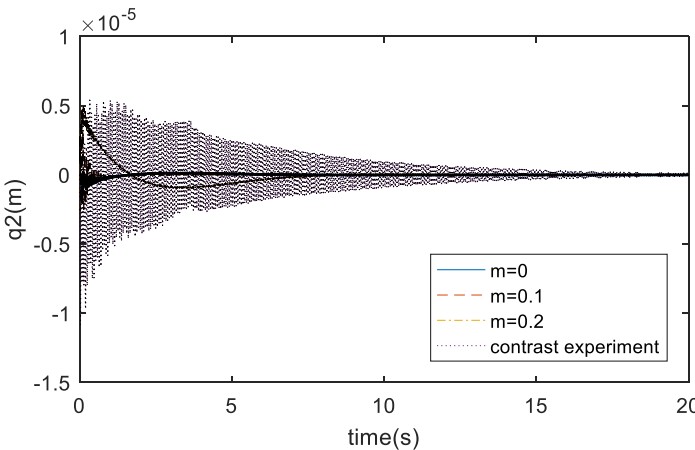

**Figure 7.** The second mode vibration of flexible manipulators.

## 5. Conclusions

In this paper, the dynamics of flexible manipulators, which are modeled as a singularly perturbed system, are decomposed into slow and fast subsystems describing the rigid and flexible motion by using the singularly perturbed theory. In the slow timescale, a slow controller based on RADP with rotating angles is designed to realize trajectory tracking. In the fast timescale, a fast controller based on RSMC is designed to suppress the vibration. The stability of the closed-loop system of tracking error dynamics is guaranteed by the proposed algorithm, and the possible numerical stiffness is also avoided. In addition, we have launched a simulation by using the proposed algorithm, in which the accuracy of the decomposition based on SP theory is proven. Moreover, the simulation results show that the composite controller is not sensitive to the varying loads, and it has better performance.

Chattering avoidance is a very important issue in the sliding-mode control, and this will be our future work with the flexible manipulator topics. On the other hand, the proposed algorithms in this paper cannot be applied to the model for a completely unknown situation, and therefore the issue of how to develop an ADP method with a singular perturbation technique with completely unknown dynamics will lead to our future research.

**Author Contributions:** Conceptualization, Y.X. and L.M.; methodology, Y.X.; software, X.W.; validation, X.W., L.W. and K.W.; formal analysis, Y.X.; investigation, X.W.; resources, Y.X.; data curation, Y.X.; writing—original draft preparation, Y.X.; writing—review and editing, Y.X. and X.W.; visualization, L.W.; supervision, K.W.; project administration, L.M.; funding acquisition, Y.X. and L.M. All authors have read and agreed to the published version of the manuscript.

**Funding:** This work is supported by the Natural Science Foundation of Jiangsu Province (Grant No. BK20200631, BK20200633, BK20200086), China Postdoctoral Science Foundation funded project 2020M681766, and National Natural Science Foundation of China (Grant No. 61873272, 62073327, 62003348).

**Conflicts of Interest:** The researchers have no conflict of interest.

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
