# Peer review of "Learning Control for Flexible Manipulators with Varying Loads: A Composite Method with Robust Adaptive Dynamic Programming and Robust Sliding Mode Control"

_electronics, doi:10.3390/electronics11060956_

Round 1
Reviewer 1 Report
The paper addresses an interesting topic. The following issues should be addressed:
- please try not to use acronyms in the title
- please improve the abstract by presenting some quantitative results
- please do not number introduction with "0"
- the contribution of the authors to section 1 is not clearly stated - reference [13] is mentioned at the beginning of the section 1 but is not clear how much of the section is taken from [13]. Please try to explain in words the structure of section 1.
- the same observation for section 2.
- section 3 is briefly presented. It is not clear which are the inputs for the provided simulations. More discussions should be provided upon results.
- concluding remarks should not include graphics. please move them into a separate discussion section and provide conclusions for the paper
- please discuss the limitations.
Author Response
Thank you very much for your positive comments, the explanation of the
modifications as well as corrections in this revision can be arranged as attacked files (comment numbers are in 1:1 correspondence with the reviewers’ comments).

Reviewer 2 Report
- The related work part should be strengthened, and some related references with nearly three years need to be supplemented.
- Insert a table to summarize the previous work with current work.
- Most of the references in this paper are too old.
- Could not trace Refs. [12,13,14,23] in the text.
- Could not trace Eq. (36) in the text.
- put a dot (.) at the end of the sentences in the citation part of all figures.
- The conclusion is not clear enough. It would be better if the author can describe it more clearly.
- In the references list, omit “Author 1; Author 2”
Author Response

(The authors gave the same response as above.)

Reviewer 3 Report
The paper content is interesting. However, the reviewer has the following recommendations:
- What is in equation 10? How did the author come up with the equation 11? Steps of developing this can be added.
- A subsection to be added to show the significance of two-time-scale feature of the flexible manipulator. It is simply mentioned in the paper, however, its significance or analytical discussion is missing.
- The main concern is the proposed controller performance has not been compared with the already proposed control performance in the literature. This comparison would provide the superiority of the proposed controller.
Author Response

(The authors gave the same response as above.)

Round 2
Reviewer 1 Report
I thank the authors for the revised version of the paper and for considering the comments from the previous round of review. In the light of the new version I have some small observations:
- please include figure 6 and 7 in the section they are mentioned and not at the end of the paper
- please include limitations and further research directions in the conclusion section
Author Response
First of all, the authors would like to express their sincere thanks to the anonymous reviewer for the helpful comments and suggestions. The explanation of the modifications as well as corrections in this revision can be arranged as follows (comment numbers are in 1:1 correspondence with the reviewers’ comments).

Reviewer 3 Report
No further comments.
Author Response
Thanks to this reviewer for the positive comments.